# Fertility Predictors in Intrauterine Insemination (IUI)

**DOI:** 10.3390/jpm13030395

**Published:** 2023-02-23

**Authors:** Anca Huniadi, Erika Bimbo-Szuhai, Mihai Botea, Ioana Zaha, Corina Beiusanu, Annamaria Pallag, Liana Stefan, Alin Bodog, Mircea Șandor, Carmen Grierosu

**Affiliations:** 1Department of Surgical Discipline, Faculty of Medicine and Pharmacy, University of Oradea, 1st December Square 10, 410073 Oradea, Romania; 2Calla-Infertility Diagnostic and Treatment Center, Constantin A. Rosetti Street, 410103 Oradea, Romania; 3Department of Pharmacy, Faculty of Medicine and Pharmacy, University of Oradea, 29 Nicolae Jiga Street, 410028 Oradea, Romania; 4Department of Clinical Discipline, Apollonia University, 700511 Iasi, Romania

**Keywords:** intrauterine, insemination, infertility, pregnancy

## Abstract

(1) Background: Intrauterine insemination (IUI) is considered a first-line procedure for infertile or hypo-fertile couples among assisted reproductive techniques. In our retrospective study, we identified variables associated with a successful IUI and the probability of obtaining a pregnancy. This is useful to identify couples with a good chance of obtaining a pregnancy through an IUI procedure (2) Methods: The study was conducted at a university-level fertility clinic in Oradea, Romania. Patients eligible to participate in the study were infertile couples who underwent IUI treatment in the interval between January 2015 and October 2020. (3) Results: In our study, we found that duration of infertility, couple age, endometrium thickness, sperm concentration, and motility are important factors in determining the outcome of IUI. Several demographics were measured for each couple including maternal and paternal age, the type and duration of infertility, the number of procedures, the type of ovarian stimulation, number of follicles, endometrial thickness, the type and day of ovulation induction, associated pathology, tubal patency, and pre/post washes sperm count and progressive sperm motility. (4) Conclusions: Fertility prediction studies are necessary, and an individualized prognostic score should be applied for each couple for tailoring their expectations and better counseling.

## 1. Introduction

Infertility is a major health issue worldwide, with an estimated prevalence of 10–15% [1]. One out of seven couples experience an unfulfilled desire to have children that last for more than a year [2]. Due to its simplicity, easy management, relatively low incidence of complications, and low cost, intrauterine insemination (IUI) is considered, among other assisted reproductive techniques, the first-line procedure for infertile or hypo-fertile couples [3,4]. Among the general indications for IUI, cervical infertility, minimal to mild endometriosis, ovulation dysfunctions, moderate male factor infertility, and also cases of unexplained sub-infertility are of significance as well [2,5]. The indications for IUI are generally related to its mechanisms of action and thought to be related to the efficacy of the treatment, which includes increasing the number of gametes (ovarian stimulation increases chances of clinical pregnancy versus the natural cycle) and increasing the probability for the gametes to be present in the female reproductive tract at the same time, therefore enhancing fertilization. In sperm, the washing procedure results in an increased number and concentration of motile sperm. In addition to improving those factors, IUI eliminates the filtering mechanism of the cervix and has been shown to increase pregnancy rates, even if semen parameters are normal [6,7].

The overall success rates of IUI are variable; the mean pregnancy rate per IUI cycle is around 9% in most of the international literature [3]. Pregnancy rates, varying from as low as 5% to as high as 70%, have been reported, and this wide range is probably because there are many contributing variables such as the heterogeneity of the population, different ovarian stimulation protocols, and the absence of well-controlled prospective randomized trails [8]. Even for specific indicators, the procedure’s effectiveness has not yet been evaluated or supported with solid data. For example, a meta-analysis has evaluated the effectiveness of IUI as a treatment for unexplained infertility in natural cycles. [9] As early as 2006, a Cochrane meta-analysis concluded that they have neither provided reliable data for a definitive comprehensive evaluation nor met the strict criteria for a definitive conclusion [9]. The factors that contribute to the success rate of an IUI are numerous and well described in the literature. Such factors include the woman’s age, the type of infertility, the number of follicles obtained by ovarian stimulation, the Estradiol concentration on the day of human chorionic gonadotrophin administration, the sperm parameters (concentration, density, and motility) the duration of infertility, endometrial thickness and the number of IUI cycles [1,4]. Knowledge of these variables and their effect on IUI is of utmost importance to gain insight into the potential success rate and to provide couples with objective counseling [5,8].

Although there are studies that recommend three to six cycles of IUI with ovarian stimulation as a first-line therapy for women younger than 40 years with unexplained infertility, we consider that IUIs are personally demanding, resource intensive, and are not without risks, one of the most prevalent risks being the frustration and demoralization of couples. In this situation, identifying the variables associated with successful IUI outcomes and the calculation of a prediction score may allow better counseling and treatment planning [10].

The predominant causes of infertility were recorded as follows: male factor infertility, female factor (minimum/mild endometriosis, anovulation or ovulation disorders including polycystic ovarian syndrome, acne, and hypothyroidism or tubal disease with only one patent Fallopian tube), and unexplained infertility [11].

Inclusion criteria refer to couples with a minimum age of 18 years, couples trying to conceive a child of at least 1 year, at least one fallopian tube must be permeable by hysterospingography or laparoscopy, absence of acute pelvic infections and normal Babes–Papanicolau cytological examination and the spermogram must have a total count of at least 5 mil/mL of spermatozoa.

Exclusion criteria included patients with bilateral tubal blockage, moderate to severe endometriosis, and severe male factor infertility (with a post-wash sperm count of <1 million/mL). However, patients with a lower sperm count after preparation on the day of the IUI procedure were included. Male factor infertility was defined as semen concentration <15 mil/mL, normal morphology <4%, and progressive motility <32% before sperm preparation [8]. For polycystic ovarian disease, patients exhibiting at least two of the following three signs were considered: anovulation, modified biochemical results, and/or ultrasound showed multiple follicles (≥10) under 10 mm diameter. In couples with unexplained infertility, the ovarian evaluation showed normal results, normal semen analysis, as well as normal ovulation and tubal patency.

## 2. Materials and Methods

### 2.1. Aim of the Study

In our study, we tried to determine which parameters could predict the success rate of an IUI procedure and determine whether couples should be counseled for an IUI or IVF (in vitro fertilization) procedure as a first-line treatment. We hypothesized that certain couple and treatment variables would predict treatment success.

Our retrospective study aimed to identify these variables associated with IUI outcomes and to develop a score to predict IUI outcomes, leading to better counseling for intrauterine insemination or in vitro fertilization.

### 2.2. Materials

This retrospective study was conducted at a university-level fertility clinic in Oradea, Romania. Patients eligible to participate in the study were infertile couples who underwent IUI treatment in the interval between January 2015 and October 2020. The clinical and biological data were collected and recorded in a medical electronic database. Each of the patients gave their written informed consent for the procedures and had their data recorded. It is of equal importance to mention that the study was approved by and by the approval of the Research Ethics Committee of the Calla-Infertility Diagnostic and Treatment Center, Oradea, Romania (no. 546/17 May 2021).

All the patients who underwent an IUI procedure in our service were eligible for inclusion throughout the study period. IUIs were performed for various causes of infertility. Infertility diagnoses were considered for all couples who had been trying to conceive a baby for at least one year. Basic investigations consisted of semen analysis, hysterosalpingography, thyroid, and prolactin determination as well as ovarian reserve markers. Patency of at least one tube, confirmed by hysterosalpingography or laparoscopy, was mandatory. All couples complying with the inclusion criteria were assessed in this research. For the analyzed period, these were 426. However, due to data quality issues, 87 of them were eliminated in the data analysis process. A final sample of 339 couples was evaluated. This is in line with our estimated required sample size.

### 2.3. Methods

For the couples who met the inclusion criteria, we registered maternal and paternal age, the complete history regarding the type of infertility and its duration, the number of procedures, the type of ovarian stimulation, number of follicles, endometrial thickness, the type of ovulation induction, day of ovulation induction, associated pathology, tubal patency and pre and post wash sperm count as well as progressive sperm motility A and B. The patients were divided into two groups: the group that achieved a pregnancy is the positive group and another group that did not achieve a pregnancy is the negative group.

### 2.4. IUI Process

Ovarian stimulation was performed with clomiphene citrate (CLOSTILBEGYT 50 mg, Egis Pharmaceuticals PLC) no 1 protocol, human menopausal gonadotrophin-(hMG) (MENOPUR 75 UI, Ferring GmbH) no 2 protocol, follitropinum alfa-rFSH (GONAL-f 75 UI Merck Europe B.V.) no 2 protocol or a combination of clomiphene citrate with hMG, no 3 protocol. The initial dose of gonadotrophin (37.5–150 UI/day) depended on the patients’ characteristics (ovarian reserve, body mass index and maternal age), was started on the 2nd day of the menstrual cycle and it was maintained for 5 days until the 6th day of the treatment and afterwards adjusted according to ovarian response. In the clomiphene and hMG group, clomiphene citrate 100 mg/day was started from the 2nd day of the menstrual cycle, and hMG 75 UI was administered on alternate days from the 5th day (on days 5, 7, 9). Once the leading follicle diameter achieved ≥17 mm, the ovulation trigger 5000 UI hCG (Ovitrelle 250 Merck Europe B.V. Amsterdam.) was administrated [8]. The IUI procedure was completed within 34–36 h after the administration of hCG. All couples were requested to abstain from intercourse for at least the day before IUI but not longer than 5 days. Semen samples were processed with the gradient method or swim-up method. The probe was stored in an incubator at 37 °C, and the IUI was performed with a soft catheter. After the procedure, all women remained in supine position for 20 min and were given luteal support with progesterone 200 mg intra vaginally 3 times daily.

The etiology of infertility for the positive group is as follows: feminine 20% (7 cases), masculine 28.6% (10 cases), mixed 14.3% (5 cases) and unexplained 37.1% (13 cases). For the negative group, the etiology was: feminine 25.7% (78 cases), 26.6% (81 cases), mixed 20.7% (63 cases) and unexplained 27% (82 cases). Without an ovarian stimulation protocol, no pregnancies resulted.

Out of the total of 35 positive patients, 28.6% (10 cases) were under stimulation with the no. 1 protocol with tablets of clomiphene citrate (CLOSTILBEGYT 50 mg, Egis Pharmaceuticals PLC), 42.9% (15 cases) were under protocol no. 2 with injections with hMG (human menopausal gonadotrophin) (MENOPUR 75 UI, Ferring GmbH, Germany), follitropinum alfa-rFSH (GONAL-f 75 UI Merck Europe B.V. Merck Europe B.V. Amsterdam), and 28.6% (10 cases) were under protocol no. 3, where we associate tablets with injections. No significant differences were shown through the analysis in the case of etiology or ovarian stimulation protocol.

### 2.5. Statistical Analysis

The data were statistically analyzed using SPSS 24. Statistical significance was considered at the standard 5% critical level. The type of statistical tests for data analysis was simple binary logistic regression for each of the significant variables, ROC (receiver operating characteristic) and the AUROC (area under the ROC), which were compared to the rank of the predictors based on the intensity of their influence. In the final step, multiple binary logistic regression was employed.

## 3. Results

### 3.1. Descriptive Analysis of the Variables Associated with Intrauterine Insemination Outcome

It is observed from Table 1 that the age of the patients is not significant at the standard 5% critical level, only at the 10% one, just like the sperm concentration post-preparation. There is a significant difference between the distributions of the two groups based on the age of the partners (*p*-value = 0.012), infertility duration (*p*-value = 0.018), endometrium thickness (*p*-value = 0.035) and semen pre-process concentration (*p*-value = 0.046). None of the other aspects considered resulted in significant differences.

Concerning etiology, 27% of females with negative outcomes and more than 37% of females with positive outcomes were initially classified as having unexplained infertility. All of the females with positive results were stimulated and treated with hMG (MENOPUR 75 UI, Ferring GmbH); follitropinum alfa-rFSH (GONAL-f 75 UI Merck Europe B.V.) was the most used (approximately 43% of this group).

### 3.2. Descriptive Analysis of the Variables in Infertility

The age of the patient, the age of the partner and the duration of infertility are minimization aspects, as the logistic regression points out. In the case of the age of the female patient, every additional year of age diminishes the chance of the woman attaining pregnancy (Exp(B) = 0.93). The same influence is given by the age of the partner; a partner 1 year older has a lower probability to contribute to a positive result in comparison with the previous year provided that all of the other aspects were identical (exp(B) = 0.906). The length of the period also negatively influences the probability of a positive result to a higher extent than both of the age variables (Table 2).

### 3.3. Descriptive Analysis of Fertility Indices

The first index is described by age and duration of infertility (age-duration), the second is described by the two sperm concentrations (concentrations pre and post-wash), and the third is described by the effects of sperm motility A and B (A_B).

From Table 3, in the analysis of the multiple logistic regression presented above, which makes up the index score (endometrial thickness, duration of infertility, sperm concentration—pre-process and post-process, sum of sperm motility percentages in the start-motility test grade A + B), results that the sperm concentration index has the highest impact in predictability, with a *p*-0.009 (*p* < 0.01 statistical significance, confidence interval—CI of 99%). The other indexes: thickness of the endometrium with a *p*-0.055 (it is a little over 0.05), and the duration of fertility in years, a *p*-0.042, have a statistical significance, CI, of 95%. The endometrium thickness is significant only at the 10% level. However, the *p*-value is very close to the 5% critical level and its coefficient is positive, showing that the odds of a positive result is 1.44 times higher for each additional millimeter of endometrial thickness. The index of age and duration is statistically significant and, as expected, its impact is negative, showing a lower probability of a positive result (pregnancy) with the increase in age and duration of infertility indices. The highest impact belongs to the sperm concentration index. Every unit increase in the concentration index increases the odds of pregnancy 1.573 times. There is no influence of sperm motility grade A and B upon the result.

## 4. Discussion

The variables studied, the age of the woman, the age of the partners, the duration of infertility, the number of IUI, the number of follicles, the thickness of the endometrium, the volume of the sperm, the concentration of the semen before and after the process, the etiology of infertility and the type of ovarian stimulation used are the important variables of IUI related to a successful outcome and which forms a score function capable of predicting IUI success.

Female age is one of the most important, and age as a significant predictor of pregnancy after assisted reproduction is well documented and has been previously reported in most studies [4]. This is because advanced maternal age decreases female fecundity, and this is due to reduced uterine receptivity and/or decreased oocyte quality [12,13,14]. Many studies have documented a significant decrease in the success rate of IUI after age 37 [15,16]. Only seven patients over 37 years of age had positive results in our study, corresponding to 20% of all positive results, i.e., 2% of all cases. Therefore, for patients aged 37 years, we consider IUI a poor treatment option [12,15,16].

The study showed that paternal age is also an important prognostic factor of a positive IUI result, with a decrease in the chances after the age of 35. Several studies have investigated the effects of paternal age on assisted reproductive outcomes with controversial results [17,18]. Recent research shows that a man’s metabolic state, including obesity and insulin resistance, plays a negative role in fertility and the future health of the newborn. The risk of these pathologies increases with age, so we believe that this could be the explanation of the impact of paternal age in the prognosis of the ART procedure (assisted reproductive techniques). Further studies including body mass index (BMI), insulin level, insulin resistance and oxidative stress markers for men should also be considered in future studies [19,20].

Duration of infertility is an important predictor of achieving pregnancy, as the success rate was significantly lower in participants with longer bouts of infertility and approximately 80% of our pregnancies were achieved after less than 4 years of trying to conceive. Although it is difficult to establish a threshold value beyond which IUI should be replaced by IVF, we believe that 4–5 years is a reasonable period of time to switch to IVF rather than IUI [21].

We found a trend toward a higher pregnancy rate with increased endometrial thickness (9.05 ± 1.23 vs. 8.59 ± 0.93, *p* value 0.035). This result is consistent with other research, although a reduction in value is difficult to establish, but most studies agree on a minimum of 7 mm [4,8].

Total motile sperm is an important prognostic factor for IUI success. We found a significantly higher pregnancy rate (18.29%) when the TMF (total mobile fraction) was in the range of 10–20 million. When considering sperm concentrations, we found that pre-trial sperm concentration was significantly different between negative group 30 mil/mL (33), *p* value 0.035 and positive group 34 mil/mL (47), *p* value 0.046. We have shown that the post-process sperm concentration is also different between the negative and the positive group, but with a *p* value of 0.072, it is considered to be statistically insignificant [21,22]. There are conflicting studies on the predictive value of progressive motility sperm, with some studies suggesting that a successful IUI requires at least 5 million motile sperm concentrations, which is a claim that is also supported by the WHO (World Health Organization) [23]. Other studies conclude that TMF has a low sensitivity for selecting couples who wish to conceive a child with IUI but a high specificity for identifying couples prone to failure of the procedure. Sperm morphology is also an important factor, and the criteria for this are controversial; since there is a lack of standardization between centers, it is necessary to establish strict criteria and reference values [24,25,26]. Testicular function and sperm quality as well as ovarian function can be influenced by a number of factors (type of nutrition, oxidative stress conditions or other). There are studies in cryogenics indicating that broccoli extract in a certain dose could improve serum and testicular oxidative biomarkers, testicular structure and function, and sperm quality before and after cryopreservation improving reproductive performance [27,28]. In addition, with regard to ovarian function, this can be reduced by using cyclophosphamide, but the simultaneous administration of N-acetylcysteine and vitamin E could significantly reduce the side effects of cyclophosphamide, especially in the ovarian tissue [29].

Limitations of our study involved a unicenter study, which included only Caucasian couples and the indications for intrauterine insemination were heterogeneous, studying both female and male causes together.

## 5. Conclusions

Fertility prediction studies are necessary, and an individualized prognostic should be applied for each couple for tailoring their expectations and better counseling. The parameters we found in our study were significantly correlated with positive IUI results, and other variables could also have a positive influence on IUI success and should be further investigated.

## Figures and Tables

**Table 1 jpm-13-00395-t001:** Variables associated with intrauterine insemination outcome.

Variable	Negative (Number of Cases)	Positive (Number of Cases)	*p*-Value
Women age	33 (7)	32 (3)	0.060
Age of the partner	35 (7)	33 (4)	0.012
Duration of infertility	4 (2)	4 (1)	0.018
IUI number	1.5 (1)	2 (2)	0.333
Follicle count	3 (1)	3 (1)	0.702
Endometrium (mm)	9 (1)	9 (2)	0.035
Volume (mL)	3 (2)	3 (2)	0.512
Semen concentration (preprocess) (%)	30 (33)	34 (47)	0.046
Semen concentration (postprocess) (%)	18 (22.8)	20 (35)	0.072
Etiology			
Female factor	78 (25.7%)	7 (20%)	0.522
Male factor	81 (26.6%)	10 (28.6%)	
Mix	63 (20.7%)	5 (14.3%)	
Unexplained infertility	82 (27%)	13 (37.1%)	
Stimulation Protocol			
None	6 (2%)	0	0.817
Clomifen citrate	81 (26.6%)	10 (28.6%)	
hMG or rFSH	120 (39.5%)	15 (42.9%)	
Protocols association	97 (31.9%)	10 (28.6%)	

**Table 2 jpm-13-00395-t002:** Univariate logistic regression and ROC curve.

Variable	Univariate Logistic Regression		ROC Curve	
	Exp (B)	*p*-Value	AUROC	*p*-Value
Female Age	0.930	0.078	0.596	0.063
Age of partner	0.906	0.015	0.630	0.012
Duration of infertility	0.735	0.023	0.619	0.021
Endometrium thickness	1.602	0.009	0.604	0.043
Concentration (pre-wash)	1.017	0.019	0.603	0.046
Concentration (post-wash)	1.028	0.005	0.593	0.072

**Table 3 jpm-13-00395-t003:** Multiple logistic regression.

Fertility Indices	B	Exp(B)	95% C.I. for Exp(B)		*p*-Value
			Lower	Upper	
Endometrium (mm)	0.364	1.439	0.992	2.088	0.055
Age—duration in couples	−0.429	0.651	0.431	0.984	0.042
Concentration	0.453	1.573	1.120	2.211	0.009
A_B total sperm motility	−0.138	0.871	0.608	1.248	0.452
Constant	−5.533	0.004			0.001

## Data Availability

Not applicable.

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
