# Peer review of "Fertility Predictors in Intrauterine Insemination (IUI)"

_jpm, 2023, doi:10.3390/jpm13030395_

Round 1

Reviewer 1 Report

The manuscript “Fertility Predictors in Intrauterine Insemination (IUI)” is aims to identify these variables associated with IUI outcomes and to develop a score to predict IUI outcomes, leading to better counseling for intrauterine insemination or in vitro fertilization.

Unfortunately, the study has serious limitans that I exposed below.
It is not clear what novel information this study provides. As was cited by the authors from their conclusions, “Fertility prediction studies are necessary and an individualized prognostic should be applied for each couple for tailoring their expectations and better counseling.”.

Each couple was treated with different protocol? How many treatments were used in the study? The couples, however, they seem to be allotted these samples randomly to three protocol treatment groups (If it is not true, description of experimental procedure might be insufficient). The reviewer thinks such experimental design is not sufficient for aim of the present study.

As indicated by the authors in Table 2, the values of area under curve (ROC) for variables such female age, concentration, or age of partner ranked from 0.593 to 0.630, which indicates low predictive capabilities of fertility.

It is stated in the Methodology section that “a final sample of 339 couples was evaluated” This seem a small number of samples, even for retrospective studies. Therefore, the selection of the samples (studies) to be analyzed may resulted in a bias in the Results and in spurious conclusions.

Thus, I am concerned that this study was insufficient in the depth of scientific research.

Author Response

The authors acknowledge the useful observations and suggestions of the reviewer’s as concerns the manuscript entitled

Fertility Predictors in Intrauterine Insemination (IUI)

Anca Huniadi1*, Erika Bimbo-Szuhai 1, Mihai Botea1, Ioana Zaha 2, Corina Beiusanu1, Annamaria Pallag 3, Liana Stefan 1, Alin Bodog1, Mircea Șandor1, Carmen Grierosu4

According to the reviewer’s recommendations, the suggestions were carefully considered, as follows:

The manuscript “Fertility Predictors in Intrauterine Insemination (IUI)” is aims to identify these variables associated with IUI outcomes and to develop a score to predict IUI outcomes, leading to better counseling for intrauterine insemination or in vitro fertilization. Unfortunately, the study has serious limitans that I exposed below.It is not clear what novel information this study provides. As was cited by the authors from their conclusions, “Fertility prediction studies are necessary and an individualized prognostic should be applied for each couple for tailoring their expectations and better counseling.”

Each couple was treated with different protocol?

The patients were treated with 3 protocols (no. 1, no. 2 and no. 3) depending on age, ovarian reserve, body mass index and previous response to treatment (simple ovarian stimulation).

 How many treatments were used in the study?

The couples, however, they seem to be allotted these samples randomly to three protocol treatment groups (If it is not true, description of experimental procedure might be insufficient). The reviewer thinks such experimental design is not sufficient for aim of the present study.

3 types of treatment protocols were used. Protocol 1 uses clomiphene citrate 50mg, Egis Pharmaceuticals PLC. Protocol number 2 uses either hMG (Menopur 75UI, Ferring GmbH) or follictropin alfa-rFSH (Gonal-f 75UI Merck Europe). Protocol number 3 uses a combination of clomiphene citrate with hMG (Menopur). The choice was not random, but depending on the particularities of the patient: age, ovarian reserve, body mass index and responses to previous stimulations.

As indicated by the authors in Table 2, the values of area under curve (ROC) for variables such female age, concentration, or age of partner ranked from 0.593 to 0.630, which indicates low predictive capabilities of fertility.

The age of the patient, the age of the partner and the duration of infertility are minimization aspects, as the logistic regression points out. In the case of the age of the female patient, every additional year of age diminishes the chance of the woman attaining pregnancy (Exp (B) = 0.93).

It is stated in the Methodology section that “a final sample of 339 couples was evaluated” This seem a small number of samples, even for retrospective studies. Therefore, the selection of the samples (studies) to be analyzed may resulted in a bias in the Results and in spurious conclusions.

There were 339 couples, 678 patients. We will consider the extremely useful observations and suggestions in the next study. Thank you very much!

There are numerous studies in the literature regarding the predictive factors of IUI success and various scoring models, but there is no agreement on them and they are not widely used.

What we are saying is that such a score should be implemented, in order to use it for individualized treatment.

Thank you very much for review reports and for the extremely useful observations and suggestions!

Kind regards,

Dr. Anca Huniadi

Reviewer 2 Report

In the present manuscript entitled " Fertility Predictors in Intrauterine Insemination (IUI)" Huniadi et al., proposed that an individualized prognostic score should be applied for each couple for tailoring their expectations and better counselling.

It is an interesting piece of work; however, I have few comments.

·       Some recent references must be included as most of the references are old.

·       The total number of patients are very less to reach out any conclusive outcome.

·       The genetic aspect of the couple must also be considered to determine the reason for infertility and inability to maintain pregnancy when artificially inseminated.

·       The role of hormones in maintaining pregnancy is well established and therefore should also be considered in designing intrauterine inseminations which are general for all types of different hormonal profile possessing population.

·       The whole manuscript need to be recheck for grammatical and syntax errors.

Author Response

The authors acknowledge the useful observations and suggestions of the reviewer’s as concerns the manuscript entitled

Fertility Predictors in Intrauterine Insemination (IUI)

Anca Huniadi1*, Erika Bimbo-Szuhai 1, Mihai Botea1, Ioana Zaha 2, Corina Beiusanu1, Annamaria Pallag 3, Liana Stefan 1, Alin Bodog1, Mircea Șandor1, Carmen Grierosu4

According to the reviewer’s recommendations, the suggestions were carefully considered, as follows:

It is an interesting piece of work; however, I have few comments.

  1. Some recent references must be included as most of the references are old.

Done

  1. The total number of patients are very less to reach out any conclusive outcome.

There were 339 couples, 678 patients. We will consider the extremely useful observations and suggestions in the next study. Thank you very much!

  1. The genetic aspect of the couple must also be considered to determine the reason for infertility and inability to maintain pregnancy when artificially inseminated.

            It is so, but due to the high cost of genetic analyses, in Romania, this type of investigation is for in         vitro fertilization.

  1. The role of hormones in maintaining pregnancy is well established and therefore should also be considered in designing intrauterine inseminations which are general for all types of different hormonal profile possessing population.

            After insemination, all patients received progestin treatment to maintain the pregnancy with          Arefam 600mg/day intravaginally, vitamins MgBe and Folic Acid, as well as low-dose Aspirin 150      mg/day.

  1. The whole manuscript need to be recheck for grammatical and syntax errors.

Done

Thank you very much for review reports and for the extremely useful observations and suggestions!

Kind regards,

Dr. Anca Huniadi

Reviewer 3 Report

Minor concerns.

In 2.4. UI Process

I'm confused with the no. 2 protocol, do you use hMG and follitropinum alfa-rFSH at the same time or is one an alternative to the other? Line 133

In line 203 it refers to the duration of fertility in years instead of infertility according to table 2,  please verify.

What Age-duration does refer in table 3 designated in fertility indices? Age partner or age female?

Author Response

The authors acknowledge the useful observations and suggestions of the reviewer’s as concerns the manuscript entitled

Fertility Predictors in Intrauterine Insemination (IUI)

Anca Huniadi1*, Erika Bimbo-Szuhai 1, Mihai Botea1, Ioana Zaha 2, Corina Beiusanu1, Annamaria Pallag 3, Liana Stefan 1, Alin Bodog1, Mircea Șandor1, Carmen Grierosu4

According to the reviewer’s recommendations, the suggestions were carefully considered, as follows:

            In 2.4. UI Process

  1. I'm confused with the no. 2 protocol, do you use hMG and follitropinum alfa-rFSH at the same time or is one an alternative to the other? Line 133

Protocol number 2 uses either hMG or alpha rFSH.

  1. In line 203 it refers to the duration of fertility in years instead of infertility according to table 2,  please verify.

Duration of infertility

  1. What Age-duration does refer in table 3 designated in fertility indices? Age partner or age female?

 Duration of infertility in couple.

Thank you very much for review reports and for the extremely useful observations and suggestions!

Kind regards,

Dr. Anca Huniadi
